# Association between serum endocan levels and organ failure in hospitalized patients with cirrhosis

**Salisa Wejnaruemarn**[1], **Sirinporn Suksawatamnuay**[1,2,3], **Jakapat Vanichanan**[4],
**Piyawat Komolmit**[1,2,3], **Sombat Treeprasertsuk**[1], **Kessarin Thanapirom**[1,2,3]*

1 Division of Gastroenterology, Department of Medicine, Faculty of Medicine, Chulalongkorn University and King Chulalongkorn Memorial Hospital, Thai Red Cross Society, Bangkok, Thailand, 2 Center of Excellence in Liver Fibrosis and Cirrhosis, Chulalongkorn University, Bangkok, Thailand, 3 Excellence Center in Liver Diseases, King Chulalongkorn Memorial Hospital, Thai Red Cross Society, Bangkok, Thailand, 4 Division of Infectious Disease, Department of Medicine, Faculty of Medicine, Chulalongkorn University and King Chulalongkorn Memorial Hospital, Thai Red Cross Society, Bangkok, Thailand

* Kessarin.t@chula.ac.th

**Data Availability Statement:** All relevant data are within the paper and its Supporting Information files.

## Abstract

### Background & aims

Acute-on-chronic liver failure is a syndrome characterized by organ failure and high short-term mortality. The lack of reliable biomarkers for the early detection of acute-on-chronic liver failure is a significant challenge. Endothelial dysfunction plays a key role in the development of organ failure. Serum endocan is a potential new biomarker for endothelial dysfunction. Therefore, this study aimed to assess the association between endocan and organ failure and 28-day mortality in patients with cirrhosis.

### Methods

Hospitalized patients with cirrhosis with and without organ failure were prospectively enrolled according to the criteria of the European Association for the Study of Liver-Chronic Liver Failure consortium. The comparative performances of serum endocan, procalcitonin, and interleukin-6 for diagnosing organ failure and predicting mortality were studied.

### Results

The study included 116 hospitalized patients with cirrhosis, 55 of whom had organ failure on admission. Patients with organ failure had significantly higher endocan, procalcitonin, and interleukin-6 levels than those without it. At a cut-off value of 15.8 ng/mL, endocan showed a sensitivity of 63.6% and specificity of 67.2% for the diagnosis of organ failure, with an area under the receiver operating characteristic curve of 0.65, which is comparable to procalcitonin and interleukin-6. Multivariate analysis identified serum endocan, creatinine, and total bilirubin as independent factors for organ failure in hospitalized patients with cirrhosis. Patients who died within 28 days had significantly higher baseline biomarker levels than those who survived. Liver failure, hospital-acquired infection, mechanical ventilator use, and interleukin-6 ≥37 pg/mL were independent predictors of 28-day mortality.

**Funding:** This study was funded by the Ratchadapiseksompotch Endowment Fund of Center of Excellence in Hepatic Fibrosis and Cirrhosis (GCE 3300170037), Ratchadapiseksompotch Fund, Faculty of Medicine, Chulalongkorn University, grant number 65/021, the Royal College of Physicians of Thailand, and the Gastroenterological Association of Thailand. The funders had no role in study design, data collection and analysis, decision to publish, or preparation of the manuscript.

**Competing interests:** The authors have declared that no competing interests exist.

## Conclusion

Serum endocan is associated with organ failure and is an independent risk factor of organ failure in hospitalized patients with cirrhosis.

## Introduction

Acute-on-chronic liver failure (ACLF) is a syndrome characterized by significant systemic inflammation and organ failure (OF) that can be acutely triggered by intrahepatic or extrahepatic insults. It carries a significant risk of short-term mortality, with 28-day and 90-day mortality rates of 33% and 51%, respectively [1–3]. OF is a major contributor to mortality, with 28-day mortality rates for one, two, three and four to six OFs of 27%, 32%, 68%, and 89%, respectively [1]. A reliable biomarker for OF is needed to help stratify and monitor treatment response in ACLF.

The presence of pathogen- and damage-associated molecular patterns, an overactive immune system, profound hemodynamic disturbances, and systemic inflammation contribute to endothelial dysfunction and OF in patients with ACLF [4]. Endothelial injury and dysfunction are often associated with increased leukocyte adhesion, increased permeability, a shift toward a procoagulant state in the hemostatic balance, and altered vasomotor tone, resulting in tissue hypoxia and, ultimately, OF. In sepsis, endothelial dysfunction is a common complication associated with OF [5]. Recent studies have shown that sepsis-induced endothelial dysfunction is a determining factor in ACLF [6] and is associated with increased mortality rates at 28 and 90 days in patients with cirrhosis [7].

Endocan, also known as endothelial cell-specific molecule-1, is a potential biomarker for endothelial dysfunction. It is a soluble 50-kDa proteoglycan produced by activated endothelial cells, mainly in the lungs and kidneys [8]. Endocan binds to leukocyte function-associated antigen-1 on the cell surface of lymphocytes, inhibiting its interactions with soluble intercellular adhesion molecule-1 and modulating leukocyte migration to inflammatory sites [9, 10]. Factors such as tumor necrosis factor alpha, interleukin-1, vascular endothelial growth factor C, and lipopolysaccharide from gram-negative bacteria can activate endothelial cells and increase endocan synthesis [9]. Previous meta-analyses have shown that high serum endocan levels are associated with various diseases, including hypertension, diabetes mellitus, chronic kidney disease, and acute respiratory distress syndrome (ARDS) [10–13]. Previous studies have suggested that serum endocan levels could be a potential biomarker for OF, survival, and severity in patients with sepsis [14–16]. Similarly, chronic liver disease is associated with endothelial cell dysfunction, and previous studies have indicated that serum endocan levels are correlated with fatty liver disease and advanced fibrosis [17]. In patients with decompensated cirrhosis, serum endocan levels were significantly different from those in the healthy controls [18]. Moreover, serum endocan can serve as an indicator of an unfavorable survival outcome in patients with cirrhosis [19]. However, there is limited research on serum endocan levels and OFs in patients with cirrhosis. This study aimed to investigate the role of serum endocan in detecting OF and predicting 28-day mortality in hospitalized patients with cirrhosis. The performance of serum endocan was also compared with that of serum procalcitonin (PCT) and interleukin-6 (IL-6) in detecting OF and predicting mortality in these patients.

## Materials and methods

### Study participants

Hospitalized patients with compensated and decompensated cirrhosis were prospectively enrolled at King Chulalongkorn Memorial Hospital in Bangkok, Thailand, between August 1, 2021 and January 31, 2023. The inclusion criteria were patients with cirrhosis aged 18 years or older who were admitted to the hospital. The diagnosis of cirrhosis was based on a combination of clinical, laboratory, and imaging evidence. The exclusion criteria were a history of hepatic or extrahepatic malignancy in the last 2 years, acute coronary artery disease, end-stage kidney disease according to the Kidney Disease Improving Global Outcomes 2012 criteria, chronic obstructive pulmonary disease, and recent antibiotic use in the previous 2 weeks.

Data were collected on clinical and laboratory characteristics. The MELD (Model for End-stage Liver Disease) [20], Child-Turcotte-Pugh (CTP) [21], Chronic Liver Failure Consortium (CLIF-C) ACLF [22], and Asian Pacific Association for the Study of the Liver ACLF Research Consortium (AARC) [23] scores were calculated upon admission and compared between the OF and non-OF groups. Blood samples were collected upon admission and centrifuged at $1,000 \times g$ at 2°C–8°C for 15 min. The supernatant was then stored at −80°C until analysis. The patients were followed up for 28 days, and deaths were recorded.

All patients or their legal guardians provided written informed consent before participating in the study. The study protocol was approved by the Institutional Review Board of the Faculty of Medicine, Chulalongkorn University (IRB No. 423/64) and was registered with the Thai Clinical Trial Registry (TCTR20220415001). The study protocol complied with the Declaration of Ethical Principles of Helsinki and the recommendations of Good Clinical Practice.

### Assessment of organ failure and bacterial infection

OF was defined according to the criteria established by the European Association for the Study of Liver-Chronic Liver Failure consortium [22]. In summary, liver failure was defined as serum total bilirubin (TB) levels of 12 mg/dL or more, kidney failure was defined as serum creatinine levels of 2 mg/dL or more or the need for renal replacement therapy, cerebral failure was defined as Grade III or IV hepatic encephalopathy according to the West-Haven criteria, coagulation failure was defined as an international normalized ratio (INR) of 2.5 or more, cardiovascular failure was defined as hypotension requiring vasopressor therapy despite adequate fluid resuscitation, and respiratory failure was defined as a ratio of arterial oxygen partial pressure to fraction of inspired oxygen of 200 or lower or pulse oximetry saturation to fraction of inspired oxygen of 214 or lower. Patients with suspected bacterial infections underwent bacterial culture for diagnosis. Bacterial infection was diagnosed using standard criteria, including clinical presentation, laboratory results, radiography, and bacterial culture [24–31].

### Measurement of serum endocan, procalcitonin, and interleukin-6 levels

Serum levels of endocan, PCT, and IL-6 were measured by enzyme-linked immunosorbent assay (ELISA) using ab278119 Human Endocan SimpleStep ELISA® Kit, ab221828 Human Procalcitonin SimpleStep ELISA® Kit, and ab178013 Human IL-6 SimpleStep ELISA® Kit (Abcam, UK), according to the manufacturer's protocol. The minimum detectable serum endocan, PCT, and IL-6 concentrations were 0.12 ng/mL, 1.51 pg/mL, and 1.6 pg/mL for serum endocan, PCT and IL-6, respectively. All analyses were performed in duplicate. Additional laboratories, including biochemistry (Abbott Laboratories, Illinois, USA), hematology (Sysmex, Kobe, Japan), and coagulation (Sysmex, Kobe, Japan), were analyzed according to the manufacturer's protocol.

## Statistical analysis

Categorical variables were presented as numbers and percentages and compared using the chi-square test or Fisher's exact test. Continuous variables were presented as mean and standard deviation or median and interquartile range (IQR) and compared using Student's *t*-test for parametric variables or the Mann-Whitney *U*-test for non-parametric variables.

The area under the receiver operating characteristic curve (AUC) was calculated to assess the performance of each biomarker in the diagnosis of OF. The AUCs were compared using the method of DeLong *et al.* [32]. The optimal cut-off value was selected based on the best sensitivity and specificity. Sensitivity, specificity, positive predictive value (PPV), and negative predictive value were calculated according to the cut-off point. Univariate and multivariate logistic regression models, including the calculation of adjusted and unadjusted odd ratios, were used to identify independent factors for OFs and predict prognosis. All statistical analyses were performed using the Statistical Package for the Social Sciences, version 28 (SPSS Inc., Chicago, Illinois, USA). A *p-value* less than 0.05 was considered statistically significant.

## Results

### Baseline characteristics

A total of 221 consecutive patients with cirrhosis were examined, 105 of whom were excluded due to the presence of cancer (*n* = 102), end-stage kidney disease (*n* = 2), and chronic obstructive pulmonary disease (*n* = 1). In total, 116 patients with cirrhosis were enrolled (Fig 1). The baseline characteristics of the study population are presented in Table 1. The mean age was 60 ± 17 years, with 71 male patients (61.2%). The most common cause of cirrhosis was alcohol-related liver disease (*n* = 35, 30.2%), followed by hepatitis B virus (HBV; *n* = 28, 24.1%), cryptogenic (*n* = 16, 13.8%), and metabolic dysfunction-associated steatotic liver disease (*n* = 15, 12.9%). The other causes of cirrhosis were hepatitis C virus (*n* = 9, 7.8%), Wilson's disease (*n* = 5, 4.3%) and others (*n* = 8, 6.9%). The CTP and MELD scores were 9 (IQR: 7–11) and 18.5 (IQR: 12–26), respectively. The most common indications for hospitalization were bacterial infection (*n* = 38, 32.8%) and gastrointestinal bleeding (*n* = 36, 31%).

At enrollment, 55 patients (47.4%) were diagnosed with OF. Of these, 34 patients (61.8%) had a single OF, 11 (20%) had two OFs, and 10 (18.2%) had three OFs. The most prevalent OFs were liver failure (52.7%), kidney failure (36.4%), and cerebral failure (30.9%). Patients with OF were younger and had higher INR, TB, creatinine, neutrophil-to-lymphocyte ratio (NLR), serum lactate, and lower sodium and albumin levels than those without OF. The median CTP (11 [IQR: 9–13] *vs.* 7 [IQR: 6–8], *p* < 0.001) and MELD (25 [IQR: 20–31] *vs.* 13 [IQR: 11–17], *p* < 0.001) were significantly higher in the OF group than in the non-OF group. The CLIF-C ACLF and AARC scores were also significantly higher in patients with OF compared to those without OF.

### Serum levels of endocan, procalcitonin, and interleukin-6 and their relationship with organ failure

Baseline levels of serum endocan (25.0 [IQR: 9.1–48.6] *vs.* 11.6 [IQR: 4.4–22.0] ng/mL, *p* = 0.002), PCT (846.5 [IQR: 338.3–2,971.5] *vs.* 263.8 [IQR: 145.0–722.0] pg/mL, *p* < 0.001), and IL-6 (60.0 [IQR: 29.1–134.9] *vs.* 18.0 [IQR: 9.2–53.9] pg/mL, *p* < 0.001) were significantly higher in patients with OF than in those without OF. The levels of all biomarkers were associated with OF severity, with higher levels associated with a greater number of OFs (S1–S3 Figs).

Regarding the type of OF, serum endocan levels were significantly higher in patients with liver, cerebral, coagulation, and cardiovascular failure (S1 Table). Serum PCT levels were

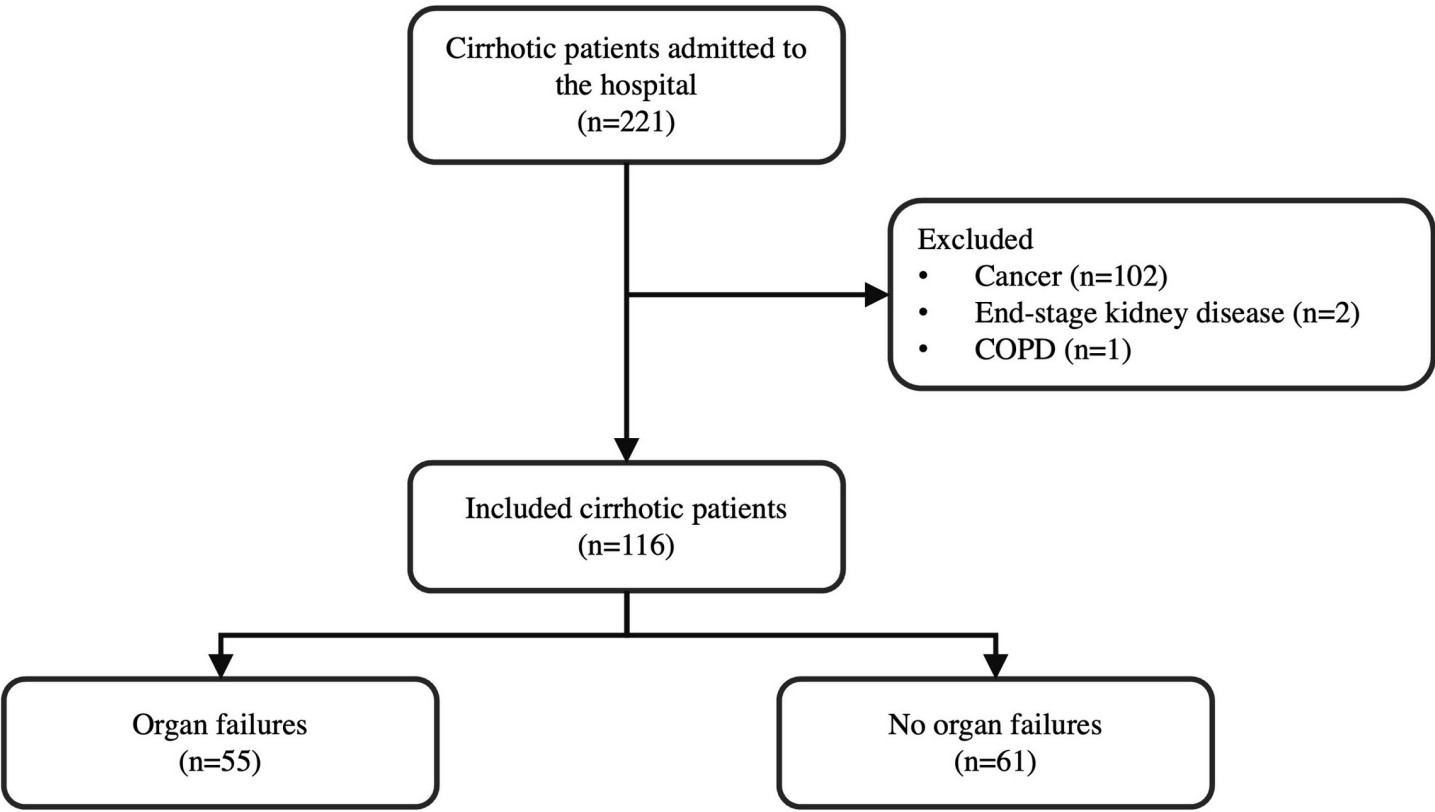

**Fig 1. Flowchart of the study protocol.** The inclusion criteria was cirrhotic patients aged 18 years or older admitted to the hospital. Patients were excluded if they had a history of hepatic or extrahepatic malignancy within the past two years, acute coronary artery disease, end-stage kidney disease, chronic obstructive pulmonary disease, or antibiotic use within the previous two weeks. Enrolled patients were classified into two groups: those with and without organ failures.

significantly higher in patients with liver, kidney, coagulation, and cardiovascular failure (S2 Table). Similar to endocan, serum IL-6 levels were significantly higher in patients with liver, cerebral, coagulation, and cardiovascular failure (S3 Table).

## Performance characteristics of serum endocan, procalcitonin, and interleukin-6 in the diagnosis of organ failure

To determine the optimal cut-off value for baseline levels of endocan, PCT, and IL-6 for the detection of OF, receiver operating characteristic (ROC) curves were generated (Fig 2). In our study of hospitalized patients with cirrhosis, the AUC was 0.65 (95% confidence interval [CI]: 0.57–0.74, $p = 0.002$) for serum endocan, 0.69 (95% CI: 0.61–0.78, $p < 0.001$) for serum PCT, and 0.67 (95% CI: 0.59–0.76, $p < 0.001$) for IL-6 in the diagnosis of OF. The AUCs of the three biomarkers were comparable for the diagnosis of OF and did not differ significantly ($p = 0.84$). The optimal cut-off value for serum endocan for the detection of OF was 15.8 ng/mL, with a sensitivity of 63.6%, specificity of 67.2%, and PPV of 63.6%. Compared with serum endocan, the performance of PCT and IL-6 in detecting OF was similar. A cut-off value of 487 pg/mL for serum PCT produced 70.9% sensitivity, 67.2% specificity, and 66.1% PPV, while a cut-off value of 37 pg/mL for serum IL-6 produced 69.1% sensitivity, 65.6% specificity, and 64.4% PPV (Table 2).

**Table 1. Patient characteristics, baseline laboratory parameters, and mortality according to organ failure.**

| Variables | Total (*n* = 116) | No OF (*n* = 61) | OF (*n* = 55) | *p* -value |
|---|---|---|---|---|
| Age (years), mean ±S.D. | 60±17 | 63±15 | 57 ±18 | 0.04 |
| Male, n (%) | 71 (61.2) | 34 (55.7) | 37 (67.3) | 0.20 |
| Presence of ascites, n (%) | 78 (67.2) | 38 (62.3) | 40 (72.7) | 0.23 |
| **Etiology of cirrhosis, n (%)** | | | | |
| Alcohol | 35 (30.2) | 20 (32.8) | 15 (27.3) | 0.52 |
| HBV | 28 (24.1) | 14 (23) | 14 (25.5) | 0.75 |
| Cryptogenic | 16 (13.8) | 7 (11.5) | 9 (16.4) | 0.37 |
| MASLD | 15 (12.9) | 9 (14.7) | 6 (10.9) | 0.54 |
| HCV | 9 (7.8) | 6 (9.8) | 3 (5.4) | 0.50 |
| Wilson's disease | 5 (4.3) | - | 5 (9.1) | 0.02 |
| Others | 8 (6.9) | 5 (8.2) | 3 (5.4) | - |
| **Comorbidity, n (%)** | | | | |
| Diabetes mellitus | 41 (35.5) | 26 (42.6) | 15 (27.3) | 0.08 |
| Hypertension | 36 (31) | 19 (31.1) | 17 (30.9) | 0.98 |
| Cardiovascular disease | 10 (8.6) | 7 (11.5) | 3 (5.5) | 0.33 |
| Chronic kidney disease | 10 (8.6) | 3 (4.9) | 7 (12.7) | 0.19 |
| No comorbidity | 39 (33.6) | 15 (24.6) | 24 (43.6) | 0.03 |
| **Indication of admission, n (%)** | | | | |
| Bacterial infection | 38 (32.8) | 19 (31.1) | 19 (34.5) | 0.70 |
| Hepatobiliary tract | 9 (7.8) | 5 (8.2) | 4 (7.3) | 0.70 |
| Urinary tract | 6 (5.2) | 4 (6.5) | 2 (3.6) | 1.00 |
| SBP | 5 (4.3) | 1 (1.6) | 4 (7.3) | 0.66 |
| Primary bacteremia | 5 (4.3) | 2 (3.3) | 3 (5.5) | 0.34 |
| Gastrointestinal tract | 4 (3.4) | 2 (3.3) | 2 (3.6) | 1.00 |
| Cellulitis | 3 (2.6) | 2 (3.3) | 1 (1.8) | 1.00 |
| Respiratory tract | 3 (2.6) | 1 (1.6) | 2 (3.6) | 1.00 |
| Unknown | 3 (2.6) | 2 (3.3) | 1 (1.8) | 1.00 |
| GI bleeding | 36 (31) | 23 (37.7) | 13 (23.6) | 0.10 |
| **Laboratory baseline (median, IQR)** | | | | |
| NLR | 5.86 (3.30–9.86) | 4.31 (2.56–8.84) | 7.27 (3.72–15.44) | 0.01 |
| Platelet ($10^3$/μL) | 121 (75–184) | 109 (70–194) | 124 (79–176) | 0.95 |
| INR | 1.50 (1.29–1.86) | 1.33 (1.23–1.53) | 1.82 (1.49–2.59) | <0.001 |
| Creatinine (mg/dL) | 1.02 (0.82–1.54) | 0.97 (0.76–1.17) | 1.16 (0.90–2.45) | <0.001 |
| Sodium (mmol/L) | 133 (129–138) | 136 (131–139) | 131 (128–136) | 0.002 |
| TB (mg/dL) | 2.63 (1.08–12.40) | 1.67 (0.82–3.68) | 13.38 (2.12–24.03) | <0.001 |
| Albumin (g/dL) | 2.8 (2.3–3.2) | 2.9 (2.5–3.6) | 2.5 (2.1–3.0) | <0.001 |
| Lactate (mmol/L) | 2.9 (1.6–4.6) | 1.9 (1.1–3.2) | 3.7 (2.6–5.8) | <0.001 |
| **Severity score** | | | | |
| CTP grade, n (%) | | | | |
| A | 19 (16.4) | 16 (26.2) | 3 (5.5) | 0.003 |
| B | 45 (38.8) | 34 (55.7) | 11 (20) | <0.001 |
| C | 52 (44.8) | 11 (18) | 41 (74.5) | <0.001 |
| CTP score | 9 (7–11) | 7 (6–9) | 11 (9–13) | <0.001 |
| MELD score | 18.5 (12–26) | 13 (11–17) | 25 (20–31) | <0.001 |
| CLIF-C ACLF score | 52 (46–58) | - | 52 (46–58) | - |
| AARC score | 9 (7–10) | 7 (6–8) | 10 (10–12) | <0.001 |
| 28-day mortality (n) (%) | 22 (19.6) | 3 (5.1) | 19 (35.8) | <0.001 |

*(Continued)*

**Table 1.** (Continued)

| Variables | Total (*n* = 116) | No OF (*n* = 61) | OF (*n* = 55) | *p* -value |
|---|---|---|---|---|
| 90-day mortality (n) (%) | 28 (32.2) | 6 (16.7) | 22 (43.1) | 0.01 |

AARC, Asian Pacific Association for the Study of the Liver (APASL) ACLF Research Consortium; CLIF-C ACLF, Chronic Liver Failure Consortium acute-on-chronic liver failure; CTP, Child-Turcotte-Pugh; GI, gastrointestinal; HBV, Hepatitis B virus; HCV, Hepatitis C virus; INR, international normalized ratio; IQR, interquartile range; MASLD, metabolic dysfunction-associated steatotic liver disease; MELD, model for end-stage liver disease; NLR, neutrophil-to-lymphocyte ratio; OF, organ failure; SBP, spontaneous bacterial peritonitis; S.D., standard deviation; TB, total bilirubin

### Factors associated with organ failure in patients with cirrhosis

A univariate analysis was performed to assess possible factors associated with OF. The presence of ascites, serum endocan, IL-6, NLR, creatinine, sodium, TB, and albumin was associated with OFs in cirrhosis. Subsequently, a multivariate logistic regression analysis was performed to examine confounding factors, including significant factors with a $p < 0.05$ identified in the univariate analysis. This included demographic data, such as age, and laboratory data, such as creatinine and TB, to identify factors associated with OF. To address multicollinearity, we removed highly correlated variables. The results indicated that serum endocan (adjusted odds ratio [aOR] = 1.02, 95% CI: 1.00–1.04, $p = 0.04$), creatinine (aOR = 5.58, 95% CI: 2.10–14.85, $p = 0.001$), and TB (aOR = 1.30, 95% CI: 1.15–1.47, $p < 0.001$) were independent factors for OFs in hospitalized patients with cirrhosis (Table 3).

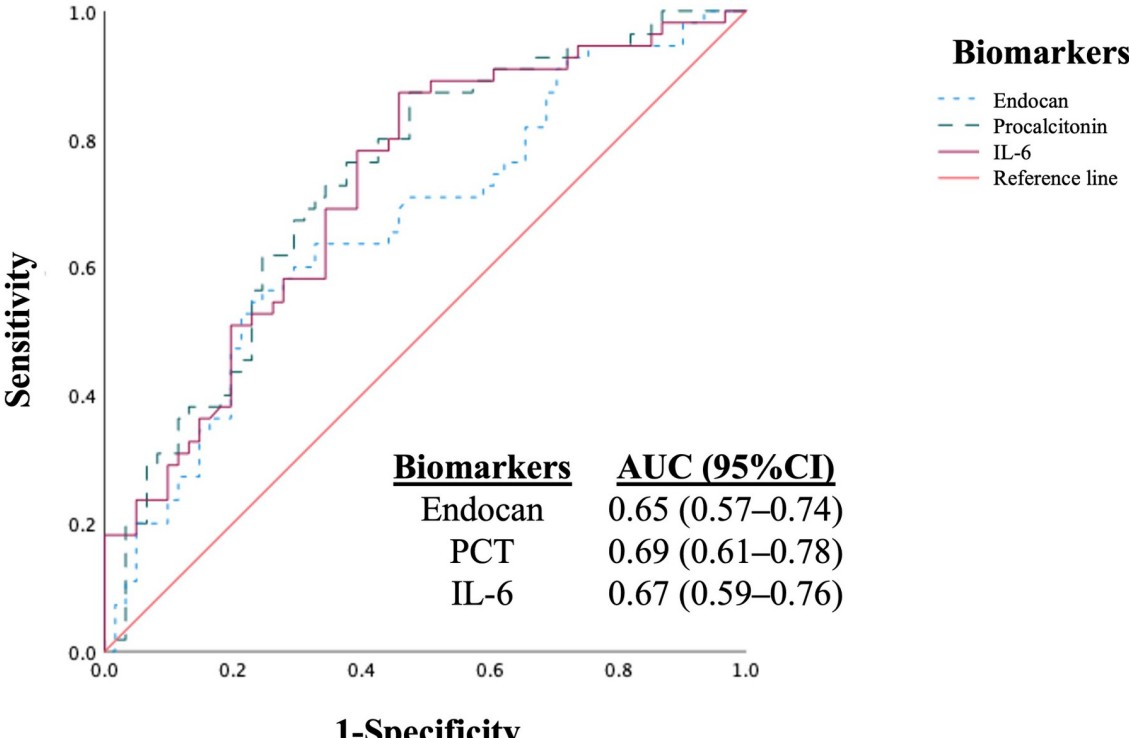

| Biomarkers | AUC (95%CI) |
|---|---|
| Endocan | 0.65 (0.57–0.74) |
| PCT | 0.69 (0.61–0.78) |
| IL-6 | 0.67 (0.59–0.76) |

**Fig 2. Performance of serum endocan, PCT, and IL-6 in the diagnosis of organ failures in hospitalized patients with cirrhosis (*n* = 116).** The AUCs for the diagnosis of organ failures were 0.65 for serum endocan, 0.69 for serum PCT, and 0.67 for serum IL-6. The AUCs of all biomarkers were not significantly different (*p* = 0.84). AUC, area under the ROC curve; IL-6, interleukin-6; PCT, procalcitonin.

**Table 2. Performance of serum biomarkers in the diagnosis of organ failure.**

| Biomarker | Cutoff | Sensitivity (%) | Specificity (%) | PPV (%) | NPV (%) | AUC (95% CI) | *p*-value |
|---|---|---|---|---|---|---|---|
| Endocan (ng/mL) | 15.8 | 63.6 | 67.2 | 63.6 | 67.2 | 0.65 (0.57–0.74) | 0.002 |
| PCT (pg/mL) | 487 | 70.9 | 67.2 | 66.1 | 71.9 | 0.69 (0.61–0.78) | <0.001 |
| IL-6 (pg/mL) | 37 | 69.1 | 65.6 | 64.4 | 70.2 | 0.67 (0.59–0.76) | <0.001 |

AUC, area under the receiver operating characteristic; CI, confidence interval; IL-6, interleukin-6; NPV, negative predictive value, PCT, procalcitonin; PPV, positive predictive value

The ROC curve was generated using a combination of independent factors associated with the OFs (Fig 3). The AUC was 0.90 (95% CI: 0.85–0.96), indicating excellent discriminative ability for identifying OF in hospitalized patients with cirrhosis.

## Serum levels of endocan, procalcitonin, and interleukin-6 and their relationship with 28-day mortality

During a median follow-up time of 113 (44–328) days, 22 patients died within 28 days (19.6%) of admission. In-hospital mortality was significantly higher in patients with an OF than in those without an OF (32.7% *vs.* 4.9%, $p < 0.001$). The most common cause of death was bacterial infection (46.4%). The characteristics and laboratory parameters of patients who died and survived for 28 days are summarized in S4 Table. Hospitalized patients with cirrhosis who died within 28 days had significantly higher baseline levels of serum endocan (28.2 [IQR: 15.1–57.2] vs 12.2 [IQR: 7.0–34.3] ng/mL, $p = 0.02$), PCT (1466.4 [IQR: 561.8–4454.9] vs 302.2 [IQR: 167.2–945.1] pg/mL, $p < 0.001$), and IL-6 (84.2 [IQR: 51.3–397.3] vs 26.8 [IQR: 11.4–60.5] pg/mL, $p < 0.001$) than those who survived.

## Predictive value of serum endocan, procalcitonin, and interleukin-6 levels in 28-day mortality

To assess the prognostic factors in hospitalized patients with cirrhosis, a univariate analysis was carried out, which included factors associated with mortality and indicated that liver failure, cerebral failure, coagulation failure, cardiovascular failure, hospital-acquired infection, mechanical ventilator use, serum endocan ≥ 15.8 ng/mL, PCT ≥ 487 pg/mL, and IL-6 ≥ 37

**Table 3. Univariate and multivariate analysis for the diagnosis of organ failures in hospitalized patients with cirrhosis.**

| Factors | Univariate analysis | | Multivariate analysis | |
|---|---|---|---|---|
| | OR (95% CI) | *p*-value | Adjusted OR (95% CI) | *p*-value |
| Age | 0.98 (0.95–0.99) | 0.05 | | |
| Ascites | 3.48 (1.57–7.73) | 0.002 | | |
| Endocan | 1.02 (1.00–1.03) | 0.02 | 1.02 (1.00–1.04) | 0.04 |
| Interleukin-6 | 1.00 (1.00–1.01) | 0.04 | | |
| NLR | 1.05 (1.00–1.10) | 0.049 | | |
| Creatinine | 3.44 (1.68–7.01) | 0.001 | 5.58 (2.10–14.85) | 0.001 |
| Sodium | 0.94 (0.88–0.99) | 0.033 | | |
| Total bilirubin | 1.23 (1.11–1.35) | <0.001 | 1.30 (1.15–1.47) | <0.001 |
| Albumin | 0.35 (0.19–0.64) | 0.001 | | |

CI, confidence interval; NLR, neutrophil-to-lymphocyte ratio; OR, odd ratio

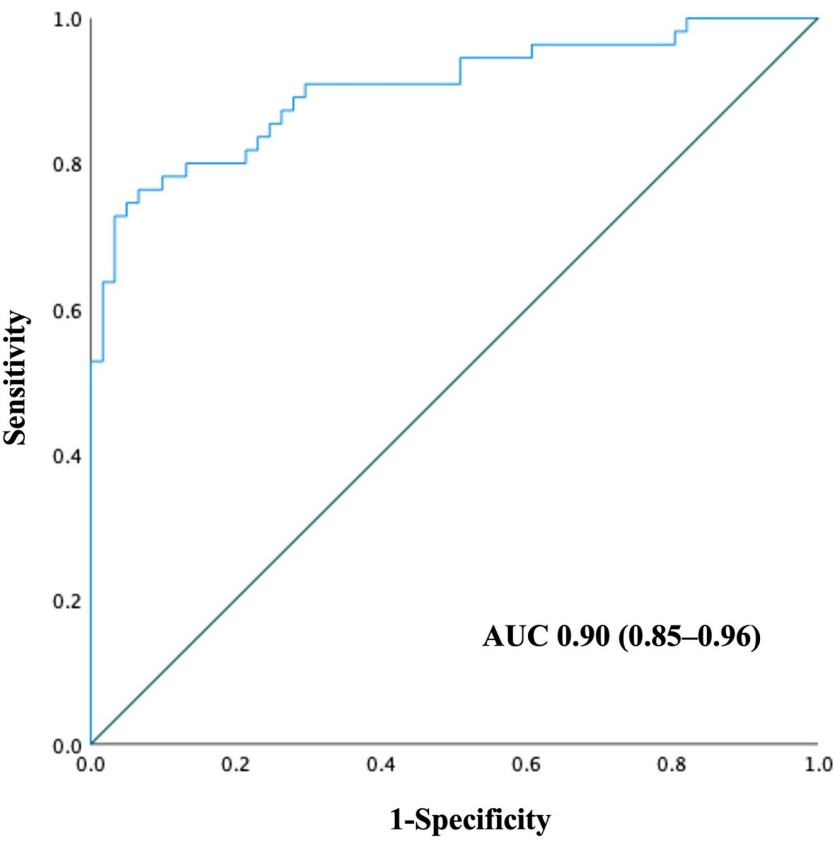

**Fig 3. Performance of the combination of serum endocan, creatinine, and total bilirubin in the diagnosis of organ failure in hospitalized patients with cirrhosis.** The AUC was 0.90 (95% CI: 0.85–0.96).

pg/mL were associated with 28-day mortality. Factors with a *p-value* less than 0.05 in the univariate analysis were included in the multivariate analysis. To address multicollinearity, we removed highly correlated variables. Subsequently, multivariate logistic regression analysis showed that liver failure (aOR = 5.23, 95% CI: 1.49–18.45, *p* = 0.01), hospital-acquired infection (aOR = 3.72, 95% CI: 1.02–13.52, *p* = 0.04), mechanical ventilator use (aOR = 7.42, 95% CI: 1.61–34.14, *p* = 0.01), and IL-6 ≥ 37 pg/mL (aOR = 20.25, 95% CI: 2.36–173.77, *p* = 0.01) were independent predictors of 28-day mortality in hospitalized patients with cirrhosis (Table 4).

The ROC curve was generated using the combination of independent factors associated with 28-day mortality from the multivariate analysis (Fig 4). The AUC was 0.90 (95% CI: 0.83–0.98), indicating excellent discriminative ability for predicting 28-day mortality in patients with cirrhosis.

## Discussion

The identification of biomarkers is fundamental for detecting ACLF in patients with cirrhosis. However, the association between serum endocan levels and OF in patients with cirrhosis has not yet been investigated. To our knowledge, this study is the first to demonstrate an association between serum endocan, a biomarker of endothelial dysfunction, and OFs in hospitalized patients with cirrhosis. In addition, the study assessed the diagnostic value of identifying OF in this population and compared its performance with that of PCT and IL-6. Our main findings

**Table 4. Univariate and multivariate analysis for the prediction of 28-day mortality in hospitalized patients with cirrhosis.**

| Factors | Univariate analysis | | Multivariate analysis | |
|---|---|---|---|---|
| | OR (95% CI) | *p*-value | Adjusted OR (95% CI) | *p*-value |
| Liver failure | 7.22 (2.62–19.92) | <0.001 | 5.23 (1.49–18.45) | 0.01 |
| Cerebral failure | 5.14 (1.70–15.58) | 0.004 | | |
| Coagulation failure | 21.50 (5.83–79.30) | <0.001 | | |
| Cardiovascular failure | 4.78 (1.09–20.91) | 0.04 | | |
| Hospital-acquired infection | 5.42 (1.92–15.27) | 0.001 | 3.72 (1.02–13.52) | 0.04 |
| Mechanical ventilation | 6.78 (2.12–21.65) | 0.001 | 7.42 (1.61–34.14) | 0.01 |
| Endocan $\geq$ 15.8 ng/mL | 4.65 (1.58–13.72) | 0.005 | | |
| Procalcitonin $\geq$ 487 pg/mL | 6.45 (2.02–20.60) | 0.002 | | |
| Interleukin-6 $\geq$ 37 pg/mL | 31.5 (4.06–244.67) | <0.001 | 20.25 (2.36–173.66) | 0.01 |
| NLR | 1.07 (1.02–1.13) | 0.01 | | |
| Albumin | 0.37 (0.17–0.79) | 0.01 | | |

CI, confidence interval; INR, international normalized ratio; NLR, neutrophil-to-lymphocyte ratio; OR, odd ratios

revealed that serum endocan levels were higher in patients with cirrhosis with OF than in those without it. A serum endocan level of 15.8 ng/ml detected OF with a sensitivity of 63.6%, specificity of 67.2%, and PPV of 63.6%. The accuracy of serum endocan in detecting OF was comparable to that of PCT and IL-6. Moreover, baseline serum endocan, TB, and creatinine

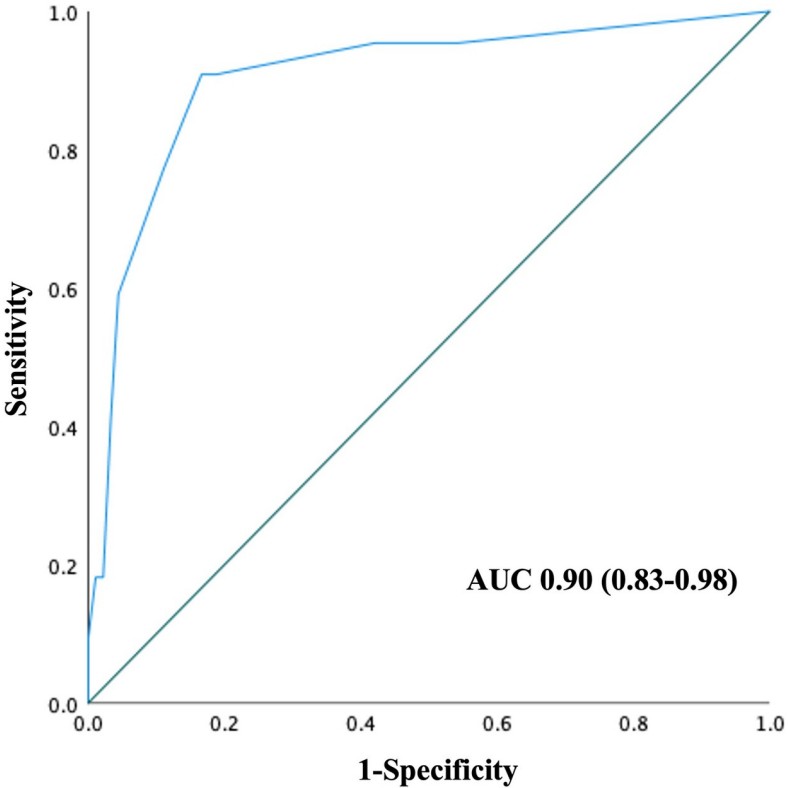

**Fig 4. Performance of the combination of liver failure, hospital acquired infection, mechanical ventilator use, and serum IL-6 with a cutoff 37 pg/mL in the prediction of 28-day mortality in hospitalized patients with cirrhosis.** The AUC was 0.90 (95% CI: 0.83–0.98). AUC, area under the ROC curve.

levels were independent factors for OFs in hospitalized patients with cirrhosis. Liver failure, hospital-acquired infection, mechanical ventilation, and IL-6 levels were independent factors associated with 28-day mortality.

Systemic inflammation plays a causal role in the development of ACLF, and vascular endothelial injury is one of the mechanisms underlying systemic inflammation-induced OF [33]. Excess proinflammatory cytokines, lipid mediators, and neutrophil migration contribute to tissue damage and endothelial dysfunction [3, 4]. Serum endocan has shown promise in the diagnosis of OF in patients with severe sepsis and septic shock. In addition, serum endocan has a prognostic ability to predict 30-day and 6-month mortality in these patients [14–16, 19]. In addition to OF in sepsis, another area of extensive research involving endocan is ARDS. A previous meta-analysis by Behnoush AH *et al*. [11] reported a correlation between serum endocan levels and ARDS, and mortality. Considering the comparable pathophysiology related to endothelial dysfunction in OF in both cirrhosis and ARDS, our study showed a similar trend in the association between serum endocan and OF, highlighting its potential as a prognostic marker. Unfortunately, the sample size of patients with respiratory failure in our study was limited ($n$ = 3), which prevented us from establishing a direct association between serum endocan levels and respiratory failure. The previous study also examined the relationship between the severity of ARDS and endocan level, but the results were conflicting. Our findings indicate a correlation between serum endocan levels and OF severity.

Our study reported that a serum endocan level of 15.8 ng/mL or higher showed fair accuracy in identifying OF (AUC = 0.65) in hospitalized patients with cirrhosis. When comparing the discriminative ability of serum endocan with that of PCT and IL-6, we found that the AUCs of serum PCT (AUC = 0.69) and IL-6 (AUC = 0.67) did not differ significantly from that of endocan, indicating that the three biomarkers had comparable efficacy in diagnosing OF in hospitalized patients with cirrhosis. Using multivariate analysis, we determined that serum endocan was an independent predictor for OFs in cirrhosis, unlike PCT or IL-6. These findings corroborate the role of endothelial dysfunction in the pathological mechanisms of OF development. Additionally, endocan is a potential biomarker for OF in patients with cirrhosis.

PCT is a precursor of calcitonin produced by parafollicular cells in the thyroid gland [34]. Its regulation is increased in bacterial infections in response to tumor necrosis factor-alpha, IL-1, and IL-6 and decreased in viral infections by interferon-gamma [35]. Several studies have demonstrated the ability of PCT to predict OF and mortality in patients with sepsis [36–38]. Furthermore, serum PCT is associated with renal dysfunction in HBV-related ACLF [39]. Zheng *et al* [39] reported higher serum PCT levels in patients with HBV-ACLF than in healthy individuals. They identified serum PCT levels of 925 pg/mL or more (87% sensitivity, 80.4% specificity, AUC = 0.84) in males and 735 pg/mL or more (77.8% sensitivity, 88.5% specificity, AUC = 0.84) in females as indicators of renal dysfunction in patients with HBV-ACLF [39]. Sato *et al*. demonstrated that serum PCT levels 50 pg/mL or more are independent prognostic factors of liver cirrhosis [40]. IL-6 is a proinflammatory cytokine linked to the progression and severity of organ dysfunction [41], and its elevated levels on day 1 had a high predictive value for early multiple organ dysfunction in critically ill patients [42] and patients with HBV-ACLF [43]. According to present study, baseline serum PCT and IL-6 levels were associated with OF. IL-6, with a cut-off point of 37 pg/mL, was an independent indicator of 28-day mortality in patients with cirrhosis. Furthermore, the combination of serum IL-6 with a cut-off point of 37 pg/mL, liver failure, hospital-acquired infection, and mechanical ventilator use showed excellent prognostic ability for predicting 28-day mortality.

The study's main strength lies in validating the diagnostic value of serum endocan, along with widely used biomarkers, such as PCT and IL-6, for detecting OF in hospitalized patients with cirrhosis. The accuracy of these biomarkers was comparable, demonstrating their

potential utility in providing physicians with an additional reliable method for identifying OFs in this population. Another notable strength of this study was that serum endocan served as an independent factor for OF and, when combined with other independent variables, produced a high AUC for detecting OF. However, our study has some limitations. First, this study did not have a validation cohort, which will determine its reproducibility and generalizability to new and different patients. Future studies are needed to confirm our findings. Second, we enrolled patients who had not had active cancer in the 2 years before recruitment, without performing additional imaging tests at the time of recruitment. Some participants may have developed new cancers during recruitment, which could affect endocan levels.

This study represents an initial step toward expanding the clinical application of serum endocan. In primary care settings, it can serve as a biomarker for physicians to identify patients with cirrhosis at risk of developing OF. The prospect for future research involves assessing its potential for predicting the development of OF in high-risk groups because its pathophysiology increases more rapidly than clinical symptoms.

## Conclusion

Serum endocan is associated with OF and is an independent factor for OF in hospitalized patients with cirrhosis. The predictive ability of serum endocan, PCT, and IL-6 were comparable in differentiating OF.

## Supporting information

**S1 Fig. Level of serum endocan according to number of organ failure.** The level of serum endocan increased as the number of organ failures increased.
(TIF)

**S2 Fig. Level of serum procalcitonin according to number of organ failure.** The level of serum procalcitonin increased as the number of organ failures increased.
(TIF)

**S3 Fig. Level of serum interleukin-6 according to number of organ failure.** The level of serum interleukin-6 increased as the number of organ failures increased.
(TIF)

**S1 Table. Level of serum endocan according to type of organ failure.**
(DOCX)

**S2 Table. Level of serum procalcitonin according to type of organ failure.**
(DOCX)

**S3 Table. Level of serum interleukin-6 according to system of organ failure.**
(DOCX)

**S4 Table. Patient characteristics and baseline laboratory parameters according to 28-day mortality ($n$ = 112).**
(DOCX)

## Acknowledgments

The authors would like to thank the Division of Gastroenterology, Department of Medicine, Faculty of Medicine, Chulalongkorn University and King Chulalongkorn Memorial Hospital, Excellence Center in Liver Diseases and Center of Excellence in Liver Fibrosis and Cirrhosis,

King Chulalongkorn Memorial Hospital for their contribution to the success and completion of this study. Moreover, we would like to thank the statistician, Chonlada Phathong, from the Division of Gastroenterology, Department of Medicine, Faculty of Medicine, Chulalongkorn University.

## Author Contributions

**Conceptualization:** Jakapat Vanichanan, Sombat Treeprasertsuk, Kessarin Thanapirom.

**Data curation:** Salisa Wejnaruemarn, Piyawat Komolmit.

**Formal analysis:** Salisa Wejnaruemarn.

**Funding acquisition:** Kessarin Thanapirom.

**Investigation:** Sirinporn Suksawatamnuay.

**Methodology:** Salisa Wejnaruemarn, Piyawat Komolmit, Kessarin Thanapirom.

**Project administration:** Kessarin Thanapirom.

**Supervision:** Sombat Treeprasertsuk.

**Writing – original draft:** Salisa Wejnaruemarn.

**Writing – review & editing:** Jakapat Vanichanan, Sombat Treeprasertsuk, Kessarin Thanapirom.

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
