## [Decision Letter · Decision Letter 0]

23 Sep 2024

PONE-D-24-36598Association of serum endocan with organ failure in hospitalized cirrhotic patientsPLOS ONE

Dear Dr. Thanapirom,

Thank you for submitting your manuscript to PLOS ONE. After careful consideration, we feel that it has merit but does not fully meet PLOS ONE’s publication criteria as it currently stands. Therefore, we invite you to submit a revised version of the manuscript that addresses the points raised during the review process.

We look forward to receiving your revised manuscript.

Kind regards,

Amir Hossein Behnoush

Academic Editor

PLOS ONE

Journal Requirements:

2. Thank you for stating the following financial disclosure: The Ratchadapiseksompotch Endowment Fund of Center of Excellence in Hepatic Fibrosis and Cirrhosis (GCE 3300170037), Ratchadapiseksompotch Fund, Faculty of Medicine, Chulalongkorn University, grant number 65/021, the Royal College of Physicians of Thailand, and the Gastroenterological Association of Thailand.

Reviewers' comments:

Reviewer's Responses to Questions

**Comments to the Author**

1. Is the manuscript technically sound, and do the data support the conclusions?

Reviewer #1: Yes

Reviewer #2: Yes

Reviewer #3: Yes

2. Has the statistical analysis been performed appropriately and rigorously? 

Reviewer #1: Yes

Reviewer #2: Yes

Reviewer #3: Yes

3. Have the authors made all data underlying the findings in their manuscript fully available?

Reviewer #1: No

Reviewer #2: Yes

Reviewer #3: No

4. Is the manuscript presented in an intelligible fashion and written in standard English?

Reviewer #1: Yes

Reviewer #2: Yes

Reviewer #3: Yes

5. Review Comments to the Author

Reviewer #1: The study titled "Association of serum endocan with organ failure in hospitalized cirrhotic patients" is well-written with appropriate methodology and interesting findings. I have some comments for improvement:

1- Define abbreviations in their first use and make sure that abbreviated forms are being used after the definition.

2- Add the strengths of your study before the limitations section.

3- Use data from previous meta-analyses investigating the role of endocan in other diseases including COVID-19, ARDS, hypertension, OSA, diabetes, CKD, etc.

4- Mention the clinical utility of your findings for a primary care physician in the discussion.

5- I found some typos and grammatical errors. A native review is required.

Reviewer #2: The authors of the current study aimed to evaluate the association between serum endocan levels and organ failure and 28-day mortality in patients with cirrhosis. The manuscript is written clearly. The results are clearly presented and justify the conclusions.

Minor remarks are as follows:

- The references for the following statement need to be provided: “The Model for End-stage Liver Disease (MELD) score, Child-Turcotte-Pugh (CTP) score, Chronic Liver Failure Consortium (CLIF-C) ACLF score, and Asian Pacific Association for the Study of the Liver ACLF Research Consortium score were calculated…”

- The Authors should provide the manufacturer’s name, town and country (in parentheses) for the biochemistry, hematology and coagulation analyzers for the determination of mentioned parameters.

- The authors are referred to some recent findings of increased serum endocan levels in fatty liver disease and liver fibrosis, also (e.g., doi: 10.2478/jomb-2019-0042).

Reviewer #3: Wejnaruemarn et al. have performed a study on the association between serum endocan and organ failure by enrolling 116 cirrhotic patients. These are my comments:

- Calculation of adjusted and unadjusted ORs should be added to the methods section. Variables for which adjustment was performed should also be added.

- A comparison should be made with a recent systematic review on the role of endocan in ARDS (doi: 10.1002/hsr2.70044).

- A paragraph highlighting the future research perspective and possible gaps in the literature should be added before the conclusion section.

- The references prior to 2010 could be updated with those after 2010 since they provide more up-to-date findings.

6. PLOS authors have the option to publish the peer review history of their article (what does this mean?). If published, this will include your full peer review and any attached files.

Reviewer #1: No

Reviewer #2: No

Reviewer #3: No

---

## [Author Response · Author response to Decision Letter 0]

10 Nov 2024

We have included all of the recommendations provided by the reviewers for the revision of the manuscript. We sincerely appreciate the insightful and valuable comments on our manuscript from the reviewers, and we are thankful for the time and effort you invested in providing feedback.

---

## [Decision Letter · Decision Letter 1]

28 Nov 2024

Association between serum endocan levels and organ failure in hospitalized patients with cirrhosis

PONE-D-24-36598R1

Dear Dr. Thanapirom,

We’re pleased to inform you that your manuscript has been judged scientifically suitable for publication and will be formally accepted for publication once it meets all outstanding technical requirements.

Kind regards,

Amir Hossein Behnoush

Academic Editor

PLOS ONE

Additional Editor Comments (optional):

Reviewers' comments:

Reviewer's Responses to Questions

**Comments to the Author**

1. If the authors have adequately addressed your comments raised in a previous round of review and you feel that this manuscript is now acceptable for publication, you may indicate that here to bypass the “Comments to the Author” section, enter your conflict of interest statement in the “Confidential to Editor” section, and submit your "Accept" recommendation.

Reviewer #1: All comments have been addressed

Reviewer #2: All comments have been addressed

2. Is the manuscript technically sound, and do the data support the conclusions?

Reviewer #1: (No Response)

Reviewer #2: Yes

3. Has the statistical analysis been performed appropriately and rigorously? 

Reviewer #1: (No Response)

Reviewer #2: Yes

4. Have the authors made all data underlying the findings in their manuscript fully available?

Reviewer #1: (No Response)

Reviewer #2: Yes

5. Is the manuscript presented in an intelligible fashion and written in standard English?

Reviewer #1: (No Response)

Reviewer #2: Yes

6. Review Comments to the Author

Reviewer #1: (No Response)

Reviewer #2: The Authors have made corrections according to the Reviewer's suggestions and improved the manuscript.

7. PLOS authors have the option to publish the peer review history of their article (what does this mean?). If published, this will include your full peer review and any attached files.

Reviewer #1: No

Reviewer #2: No

---

## [Editor Report · Acceptance letter]

11 Dec 2024

PONE-D-24-36598R1 

PLOS ONE

Dear Dr. Thanapirom, 

I'm pleased to inform you that your manuscript has been deemed suitable for publication in PLOS ONE. Congratulations! Your manuscript is now being handed over to our production team.

Kind regards, 

on behalf of

Dr. Amir Hossein Behnoush 

Academic Editor

PLOS ONE